# Profound alteration in cutaneous primary afferent activity produced by inflammatory mediators

Kristen M Smith-Edwards[1,2], Jennifer J DeBerry[3,4,5], Jami L Saloman[3,4,5], Brian M Davis[3,4,5]*, C Jeffery Woodbury[1]*

[1]Department of Zoology and Physiology, University of Wyoming, Laramie, United States; [2]University of Wyoming Graduate Program in Neuroscience, Laramie, United States; [3]Department of Neurobiology, University of Pittsburgh, Pittsburgh, United States; [4]Center for Neuroscience at the University of Pittsburgh, Pittsburgh, United States; [5]Pittsburgh Center for Pain Research, Pittsburgh, United States

**Abstract** Inflammatory pain is thought to arise from increased transmission from nociceptors and recruitment of 'silent' afferents. To evaluate inflammation-induced changes, mice expressing GCaMP3 in cutaneous sensory neurons were generated and neuronal responses to mechanical stimulation in vivo before and after subcutaneous infusion of an 'inflammatory soup' (IS) were imaged in an unanesthetized preparation. Infusion of IS rapidly altered mechanical responsiveness in the majority of neurons. Surprisingly, more cells lost, rather than gained, sensitivity and 'silent' afferents that were mechanically insensitive and gained mechanosensitivity after IS exposure were rare. However, the number of formerly 'silent' afferents that became mechanosensitive was increased five fold when the skin was heated briefly prior to infusion of IS. These findings suggest that pain arising from inflamed skin reflects a dramatic shift in the balance of sensory input, where gains and losses in neuronal populations results in novel output that is ultimately interpreted by the CNS as pain.

*For correspondence: bmd1@ pitt.edu (BMD); woodbury@uwyo. edu (CJW)

**Competing interests:** The authors declare that no competing interests exist.

## Introduction

Increased pain from stimulation of inflamed tissues is generally thought to arise from nociceptors that have become more responsive to mechanical stimuli as a result of exposure to inflammatory mediators released at the site of injury. In addition, inflammation may also recruit 'silent' afferents that are normally unresponsive to mechanical stimuli but gain mechanical sensitivity de novo in the presence of chemical mediators, providing novel input to pain pathways (*Davis et al., 1993*; *Feng and Gebhart, 2011*; *Habler et al., 1990*; *Meyer et al., 1991*; *Neugebauer et al., 1989*; *Schmelz et al., 1994*; *Schmidt et al., 1995*; *Xu et al., 2000*). Through positive feedback, the enhanced transmission of sensory information from both populations triggers a host of peripheral and central changes that can lead to chronic pain (*Gold and Gebhart, 2010*; *Koltzenburg, 1995*; *McMahon and Koltzenburg, 1990*).

The sensitization of nociceptors by inflammatory mediators has been well documented (*Bevan and Yeats, 1991*; *Fjallbrant and Iggo, 1961*; *Randich et al., 1997*; *Grigg et al., 1986*; *Schaible and Schmidt, 1988*; *Steen et al., 1992*; *Steranka et al., 1988*). Nevertheless, many questions remain with regard to 'silent' afferents, including their frequency of occurrence, whether activation by inflammatory mediators initiates their unmasking, and how quickly this unmasking can occur. Previous studies based on single cell electrophysiological approaches have been limited in their ability to assess the development of sensitization within a population of neurons due to spatial and

temporal restrictions. These limitations may explain the conflicting results regarding peripheral sensitization of cutaneous afferents to mechanical stimuli, as well as the disagreement in the relative proportions of 'silent' afferent populations in various tissues (*Lynn, 1991*; *Michaelis et al., 1996*). By contrast, the use of genetically encoded calcium indicators, such as members of the GCaMP family of molecules, provides the unique ability to simultaneously monitor the activity of a population of cells in real time (*Akerboom et al., 2009*; *Park and Dunlap, 1998*; *Tian et al., 2009*). In living cells, the fluorescent signal generated by GCaMP molecules is determined by the level of GCaMP expression and the calcium concentration within a cell that dynamically regulates the three-dimensional conformations of the GCaMP molecule and its green fluorescent protein (GFP) light emission characteristics. Therefore, GCaMP allows non-invasive, quantitative analyses of neural activity over time and is an ideal approach to detect changes in mechanical sensitivity across primary afferent populations in vivo.

Here, the generation and characterization of a transgenic mouse line that ubiquitously expresses the genetically encoded calcium indicator, GCaMP3 (third generation GCaMP), in dorsal root ganglion (DRG) neurons is described. Expression of GCaMP3 enabled the observation of calcium transients in the somata of A- and C-fiber afferents, in vivo and ex vivo, primarily in neurons with broad somal action potentials, indicating that these represented populations of nociceptors. To determine the effects of a cocktail of inflammatory mediators on the mechanical sensitivity of a large population of sensory neurons, mechanically evoked intracellular calcium transients in cutaneous primary afferents were optically recorded in an unanesthetized, in vivo preparation before and after a subcutaneous infusion of 'inflammatory soup' (IS). IS caused some primary afferents to acquire sensitivity to mechanical stimulation de novo and the number of such afferents increased five fold if the IS was preceded with 3 s of noxious heat stimulation; these neurons fit the definition of 'silent' afferents. Moreover, IS infusion altered the sensitivity of the majority of mechanically sensitive afferents. Some displayed increased calcium responses to mechanical stimulation; however, a surprisingly large population of neurons showed decreased responses to mechanical stimulation or stopped responding altogether after IS. This indicates that acute inflammation causes a profound imbalance of sensory input to the spinal cord compared to normal conditions.

## Results

### GCaMP3 immunofluorescence and calcium signaling in DRG neurons in vitro

To facilitate detection of emergent activity in diverse and potentially rare cell populations, a mouse line was generated that expressed GCaMP3 ubiquitously across all neurons by crossing Ai38 mice containing a *loxP*-flanked GCaMP3 construct within the *Gt(ROSA)26Sor* locus (The Jackson Laboratory, catalogue #014538) with mice that express Cre recombinase under the EIIa adenovirus promoter (The Jackson Laboratory, catalogue #003724); the latter strain expresses Cre recombinase in the embryo prior to implantation, producing a mosaic pattern that often includes germ cells. Mice were bred until germline expression of GCaMP3 was achieved as determined by 100% transmission of GCaMP3 to all offspring from crosses involving one GCaMP3-positive male and wild-type females.

DRG neurons in paraformaldehyde-fixed sections exhibited variable levels of native GCaMP3 fluorescence (*Figure 1A*). This variable GCaMP3 signal raised the question as to whether there was variable expression of GCaMP3 protein in different sensory neurons. Immunostaining with an anti-GFP antibody revealed that virtually all DRG neurons exhibited a detectable level of GCaMP3-like staining that was at least five standard deviations above background (*Figure 1B,C*); individual satellite or endothelial cells could not be distinguished. As with native GCaMP3 signal, the level of the immunofluorescent signal was variable, with small somata giving the brightest signal.

To determine whether GCaMP3 was expressed at sufficient levels to report neuronal activity, freshly excised DRG neurons were enzymatically treated to facilitate dissociation, plated on coverslips, and imaged during depolarization evoked by brief application of 50 mM $K^+$. Prior to $K^+$ exposure, there was a wide range of baseline GCaMP3 signals similar to that seen in fixed tissue. Application of $K^+$ produced a rapid and robust calcium transient, measured by a change in GCaMP3 fluorescence (F), in the vast majority of neurons (462/474, 97.5%; mean $\Delta F/F_0 = 1.61 \pm 0.03$). In

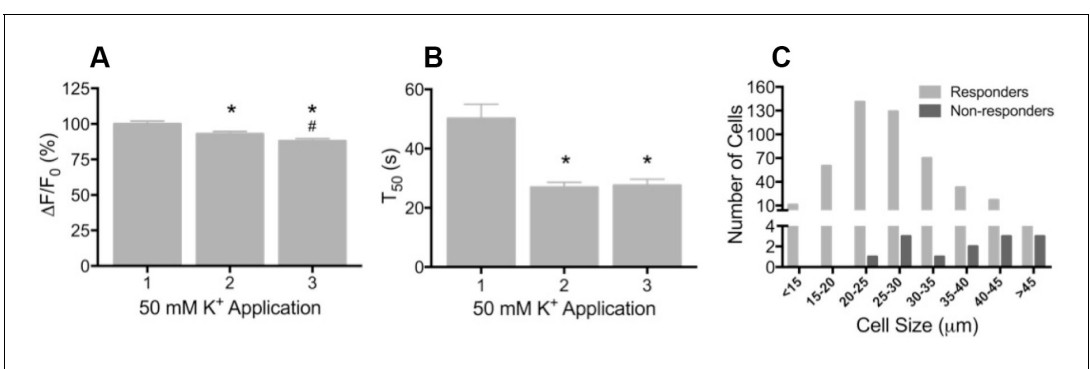

**Figure 1.** DRG neurons exhibit variable levels of native GCaMP3 fluorescence. Paraformaldehyde-fixed sections of GCaMP3-expressing DRG. (**A**) Endogenous GCaMP3 signal demonstrated a wide range of resting GCaMP3 fluorescence. Arrow indicates cell with a small somata and high resting GCaMP3 signal. Arrowhead indicates somata with low GCaMP3 signal. Asterisk indicates cell with large somata and no GCaMP3 signal. (**B**) The same section as in (**A**) but stained with anti-GFP antibody to boost GCaMP3 signal. Now, somata with low or no endogenous GCaMP3 signal can be seen to express the transgenic GCaMP3 protein. (**C**) Merged images. Scale bar, 50 µm.

neurons that exhibited a change in GCaMP3 signal, the peak amplitude of evoked calcium transients significantly decreased with repeated (×3) application of $K^+$ (mean $\Delta F/F_0$ as a percentage of $K^+$ #1: $K^+$ #1 = 100 ± 2.07; $K^+$ #2 = 92.95 ± 1.82; $K^+$ #3 = 87.98 ± 1.75; p<0.0001) (**Figure 2A**). The decay time ($T_{50}$) of evoked transients, measured in 274/474 neurons that returned to baseline after $K^+$ #1, also significantly decreased with repeated $K^+$ application (p<0.0001; **Figure 2B**). The decreases in amplitude and decay most likely reflect an engagement of intracellular calcium buffering mechanisms as a result of the initial depolarization (*Berridge, 2003*; *Berridge et al., 2003*). A small subset of neurons (12/474; 2.5%) with large diameters exhibited no detectable transient (**Figure 2C**). In vivo, the percent of neurons that did not exhibit a GCaMP3 signal is probably higher than the 2.5% reported here; dissociation and culture of DRG neurons is accompanied by a loss of up to 50% of all cells in the ganglia and neurons with large diameters are particularly susceptible to death and/or loss during processing (*Malin et al., 2007*).

## GCaMP3 signal in DRG ex vivo

To evaluate the reliability of GCaMP3 to report individual action potentials in DRG neurons in situ, controlled electrical stimuli were delivered to the dorsal cutaneous nerve (DCN) and/or spinal (i.e., intercostal) nerve of intact thoracic ganglia in ex vivo preparations (n=4) using suction electrodes. A

**Figure 2.** Depolarization evokes reproducible GCaMP3 signals in vitro (**A**) Application of 50 mM $K^+$ produced a robust fluorescent signal in the vast majority of dissociated DRG neurons, and peak evoked fluorescence ($F–F_0$) expressed as a% of the initial $K^+$ application significantly decreased with subsequent applications. Data are presented as mean ± SEM. *p<0.001 versus $K^+$ #1; #p<0.0001 versus $K^+$ #2. (**B**) The time to decay to 50% of peak signal ($T_{50}$) following $K^+$-evoked depolarization also significantly decreased with repeated application of $K^+$. Data are presented as mean ± SEM. *p<0.0001. (**C**) Distribution of somata diameter of cultured DRG neurons that did (light gray bars) and did not (dark gray bars) exhibit an increase in GCaMP3 signal in response to application of 50 mM $K^+$. Responders were small-to-medium in size (median 25–29 µm), and cells that did not exhibit a $K^+$-evoked GCaMP3 signal (non-responders) were ≥30 µm in size (median 45–49 µm). Data are presented in 5 µm bins. N=5.

large subset of DRG neurons, that included both A- and C-fiber afferents, exhibited clearly detectable phase-locked GCaMP3 signals that slowly summated in response to stimulation at 1 Hz (*Figure 3A*). Higher stimulation frequencies revealed additional cells, although individual spikes could not be resolved above 10 Hz (*Figure 3A*). In most cases, the greatest recruitment of cells was seen between 10–20 Hz, with some cells exhibiting a decrease in the intensity of the GCaMP3 response at 100 Hz, presumably reflecting their inability to reliably follow this stimulation frequency. Most GCaMP3-responding cells appeared to have small- to medium- sized somata and exhibited baseline fluorescence in the absence of stimulation, paralleling observations in fixed tissue sections (described above). All responding cells that were examined with intracellular recording techniques had broad, inflected somal action potentials (APs) and peripheral conduction velocities (CVs) in either the C- (n=10) or Aδ-fiber (n=3) range (*Figure 3B,C*).

Interestingly, electrical stimulation failed to elicit a detectable GCaMP3 signal in some medium- to large-sized cells regardless of stimulation frequency or intensity, even when stimuli were applied to dorsal roots instead of peripheral nerves. This lack of GCaMP3 signaling was consistently seen across every DRG in all animals examined and was foreshadowed by findings in dissociated cells (described above). On closer inspection, it was noted that most of these non-responders also lacked baseline fluorescence in the absence of stimulation, unlike GCaMP3-responding neurons.

Many medium- to large-sized DRG neurons exhibit narrow APs that have been proposed to reduce calcium entry (*Koerber and Mendell, 1992*; *Lu et al., 2006*), which may account for the lack of GCaMP3 signal. To investigate this possibility, non-responders were targeted with intracellular recordings in some experiments (*Figure 4A*). Notably, these studies revealed that GCaMP non-

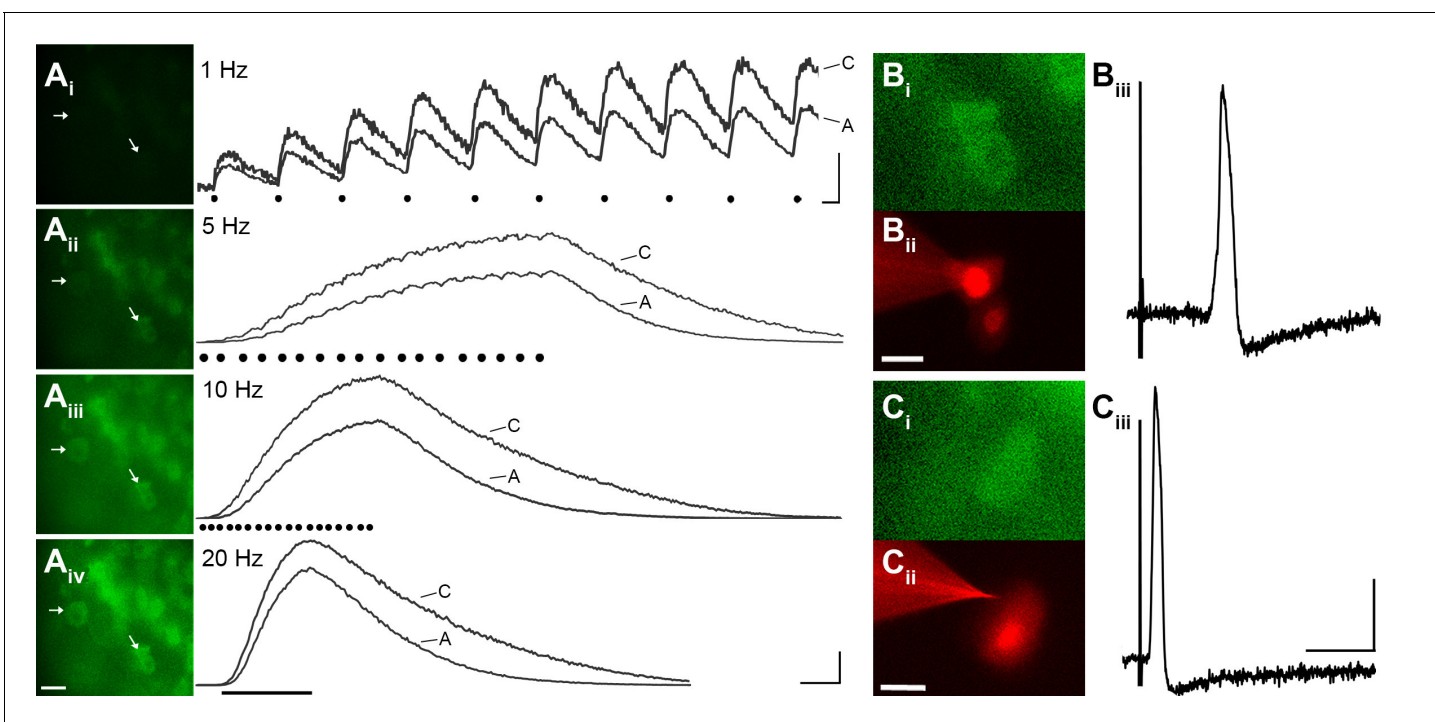

**Figure 3.** GCaMP3 reports electrically evoked spike activity from Aδ- and C-fiber afferents in the ex vivo skin-nerve preparation. (**A**) Continuous calcium transients imaged during application of electrical stimulation to spinal nerve. As electrical stimulation frequency increased (1, 5, 10, and 20 Hz; A$_i$–A$_{iv}$), the GCaMP3 signal from DRG somata increased in fluorescence intensity (images on left; scale bar, 40 µm). Right side of panel A shows GCaMP3 traces from physiologically identified cells (white arrows). The GCaMP3 signal from both the C fiber (top trace) and Aδ fiber (bottom trace) resolved single spike activity up to 10 Hz. Electrical stimuli are represented as black dots beneath traces (they fuse at 20 Hz). Scale bar, 250 ms; 5 ΔF/F$_0$ for 1 Hz; 20 ΔF/F$_0$ for 5, 10 and 20 Hz. (**B–C**) To verify that the GCaMP3 signal was reporting spike activity, intracellular recordings were also made from GCaMP3-responding afferents. GCaMP3-responding cells were located (B$_i$ and C$_i$) and impaled with electrodes containing AlexaFluor-555 (red) to confirm the identity of cells (B$_{ii}$; C$_{ii}$; scale bar, 20 µm). Representative action potentials from a C fiber (B$_{iii}$; conduction velocity, CV = 0.51 m/s$^2$) and Aδ fiber (C$_{iii}$; CV = 2.2 m/s$^2$) are shown on the right. Scale bar, 10 ms; 20 mV.

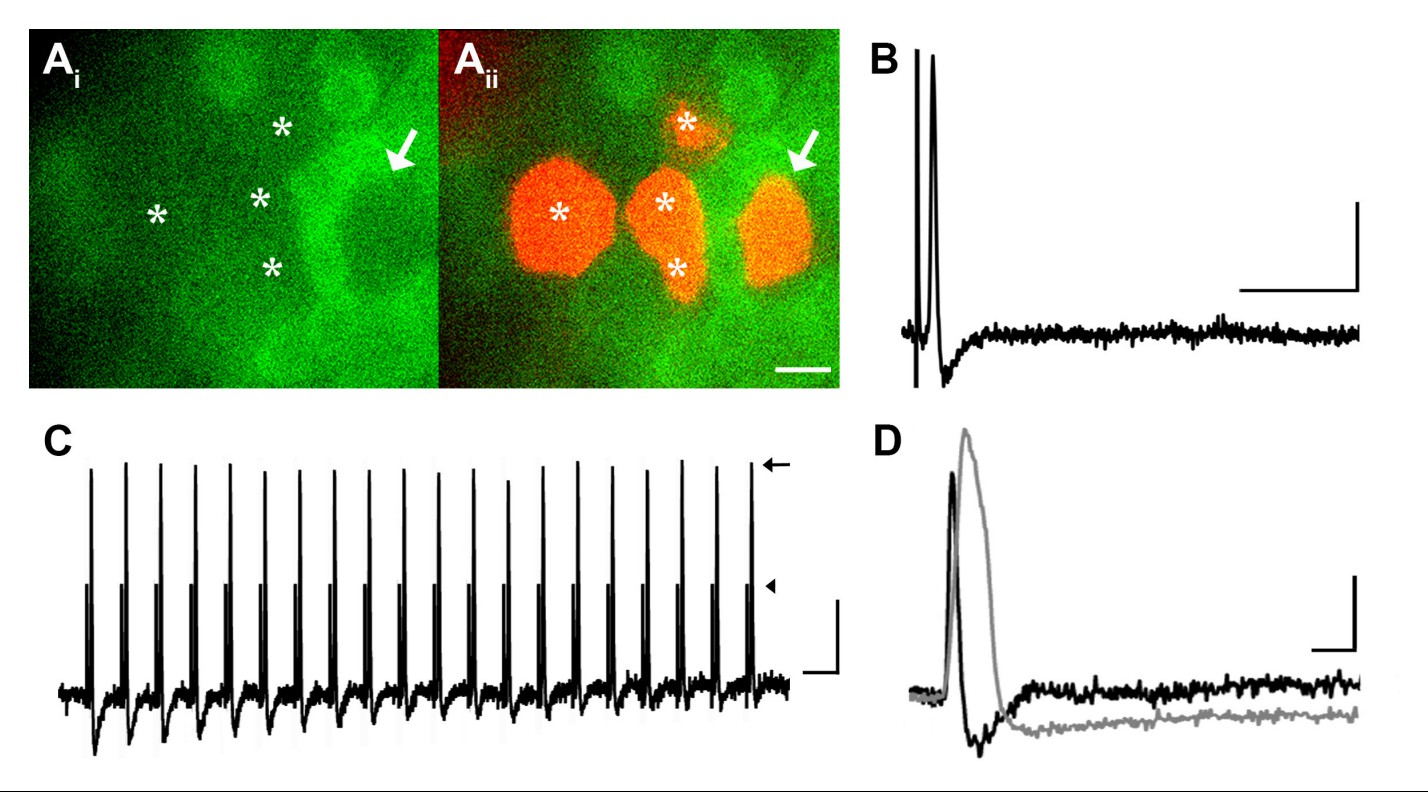

**Figure 4.** DRG neurons with narrow spikes and fast conduction velocities do not exhibit GCaMP3 signals, despite being electrically driven in the ex vivo preparation. (A) Spinal nerve branches (dorsal cutaneous and intercostal nerves) were electrically stimulated concurrently at 20 Hz (found to maximally elicit a GCaMP3 response) with suction electrodes. Not all DRG somata exhibited a GCaMP3 signal (A$_i$; GCaMP3-non-responders indicated by white asterisks and white arrow). These afferents were impaled for intracellular recording and their identity was verified with AlexaFluor-555 (A$_{ii}$). Scale bar, 20 μm. (B) Intracellular recording confirmed that GCaMP3-non-responding afferents were firing action potentials with CVs in Aβ- and Aδ-fiber ranges (example action potential from neuron indicated by arrow in A; CV = 8.2 m/s$^2$). Scale bar, 10 ms; 20 mV. (C) GCaMP3-non-responding afferents followed stimulation at 100 Hz, but still did not exhibit a GCaMP3 signal. The stimulus artifacts have been cropped (at arrowhead) to better visualize action potentials (arrow). Scale bar, 1 ms; 20 mV. (D) Superimposed action potentials from a GCaMP3-non-responding (black) and a GCaMP3-responding (gray) myelinated afferent reveal that GCaMP3 responders have broad spikes, whereas GCaMP3 non-responders have narrow, uninflected spikes. Scale bar, 2 ms; 20 mV.

responders reliably follow high frequency trains of stimulation with high fidelity and exhibited narrow, uninflected APs (8/8, *Figure 4B*). Surprisingly, even driving these cells with a train of spikes at 100 to 200 Hz failed to produce a GCaMP3 signal (*Figure 4C*). Compared to GCaMP3-responding neurons, non-responders had significantly narrower APs (0.85 ± 0.12 ms vs. 2.5 ± 0.25 ms; p<0.001) (*Figure 4D*), larger somata (34.1 ± 1.1 μm vs. 23.6 ± 1.3 μm; p<0.001), and faster CVs (5.3 ± 1.6 m/s vs. 0.92 ± 0.29 m/s; p=0.031). Because most DRG cells exhibiting narrow, uninflected somal spikes at 30–31°C are either tactile or proprioceptive afferents (*Boada and Woodbury, 2007*; *Malin et al., 2007*), these data suggest that GCaMP3 may not be optimal for the study of fast conducting, low-threshold, DRG neurons.

One caveat to these conclusions comes from the report by Lu et al. (*Lu et al., 2006*) that found whereas small diameter sensory neurons exhibited large transients (imaged with Fura-2) similar to the results here, cells with large diameter somata had small, but detectable calcium transients. The difference between observing small transients in large cells (Lu et al.) and no signal at all (the present report) could in part be due to the higher $K_d$ for $Ca^{2+}$ of GCaMP3 relative to Fura-2 (350 nM vs. 140 nM). However, in transgenic mice expressing GCaMP6s (that has a similar $K_d$ to Fura-2 [144 nM]), no signal could be detected in many large diameter DRG neurons during dorsal root stimulation at 100 Hz (ex vivo preparations) or in spindle afferents during maintained muscle stretch (in vivo preparations; Smith-Edwards and Woodbury, unpublished data), another population of large

cells with narrow spikes. It should also be noted that the absence of calcium transients was rare in dissociated neurons in the present report (seen in only 2.5% of neurons *Figure 2*) and thus, does not represent a major conflict between the two reports.

## GCaMP3 signal in DRG in vivo

The utility of GCaMP3 for studies of sensory neuron activity in vivo was initially evaluated by imaging lumbosacral DRG during application of electrical stimuli to peripheral targets (n=7 mice). As in fixed, dissociated, and intact ex vivo DRG (described above), baseline GCaMP3 fluorescence was evident in many neurons in the absence of stimulation. Similar to ex vivo preparations, electrical stimulation through bipolar electrodes inserted at the base of the tail revealed that individual phase- locked fluorescent transients could be detected at 1 Hz in vivo (*Figure 5*).

The GCaMP3 signal from individual spikes tended to be slightly noisier in vivo than ex vivo due to movement artifact during imaging, but at lower stimulation frequencies (e.g., 0.5 Hz) the signal intensities from individual spikes were found to remain remarkably stable over time, with no obvious

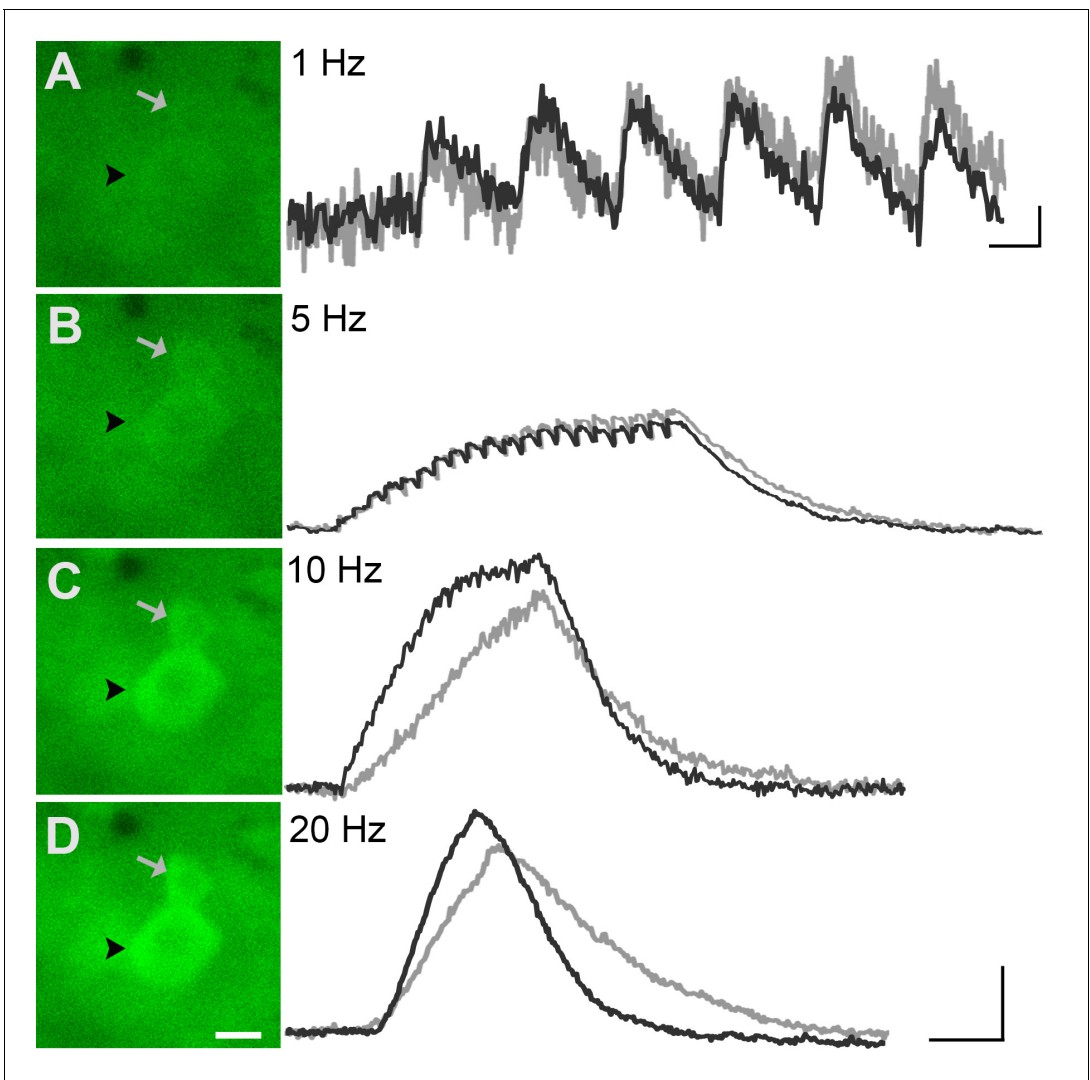

**Figure 5.** GCaMP3 fluorescence reports electrically evoked activity from DRG afferents in vivo. (**A–D**) The GCaMP3 signal from two DRG somata (black arrowhead and gray arrow) in response to 1, 5, 10, and 20 Hz electrical stimulation of receptive field (from top to bottom) was imaged (shown on left; scale bar, 20 μm). GCaMP3 traces (shown on right) from the identified cells reveal that single spikes can be resolved up to 10 Hz, similar to findings from the ex vivo preparation. Scale bars, 500 ms; 5 ΔF/F$_0$ for 1 Hz, and 10 ΔF/F$_0$ for 5, 10 and 20 Hz.

change in peak intensity when examined at 30-min intervals (average percent change in response at 30 min was −0.082 ± 0.15, p=0.53; n=2 mice, 5 cells). Also as in ex vivo preparations, signal intensity increased with higher stimulation frequency but again, individual APs could not be resolved above 10 Hz (*Figure 5*). Unlike ex vivo findings, however, electrically evoked GCaMP3 signals in DRG somata showed a less-than-monotonic increase with increasing frequencies of electrical stimulation, presumably reflecting reduced fidelity of activation with bipolar vs. suction electrodes.

Next, the ability of GCaMP3 signaling to reliably report variations in the intensity of an ascending series of controlled mechanical forces to the skin was assessed in order to address the utility of GCaMP3 for visualizing changes in physiologically relevant activity of sensory neurons over time (i.e., plasticity). As shown in *Figure 6A and B*, mechanically-evoked fluorescent signals in small- and medium-sized somata were proportional in amplitude to stimulus intensity, a finding that remained stable upon repeated stimulation at 30-min intervals for up to 90 min (*Figure 6C,D*; average percent change in response across 32 cells in 3 mice = –0.026 ± 0.054, p=0.754).

## Inflammation-induced plasticity

In light of the observed stability in GCaMP3 signaling in response to repeated mechanical stimulation, GCaMP3-expressing mice were then used to examine nociceptor plasticity in real time. In particular, one of the long-standing issues in the development of inflammatory pain has been the role of 'silent' nociceptors (*Gold and Gebhart, 2010*; *Koltzenburg, 1995*; *McMahon and Koltzenburg, 1990*). Such chemosensitive afferents have been deduced to constitute up to 30% of all sensory neurons in the DRG (*Davis et al., 1993*; *Feng and Gebhart, 2011*; *Habler et al., 1990*; *Kress et al., 1992*; *Meyer et al., 1991*; *Neugebauer et al., 1989*; *Schmelz et al., 1994*; *Schmidt et al., 1995*;

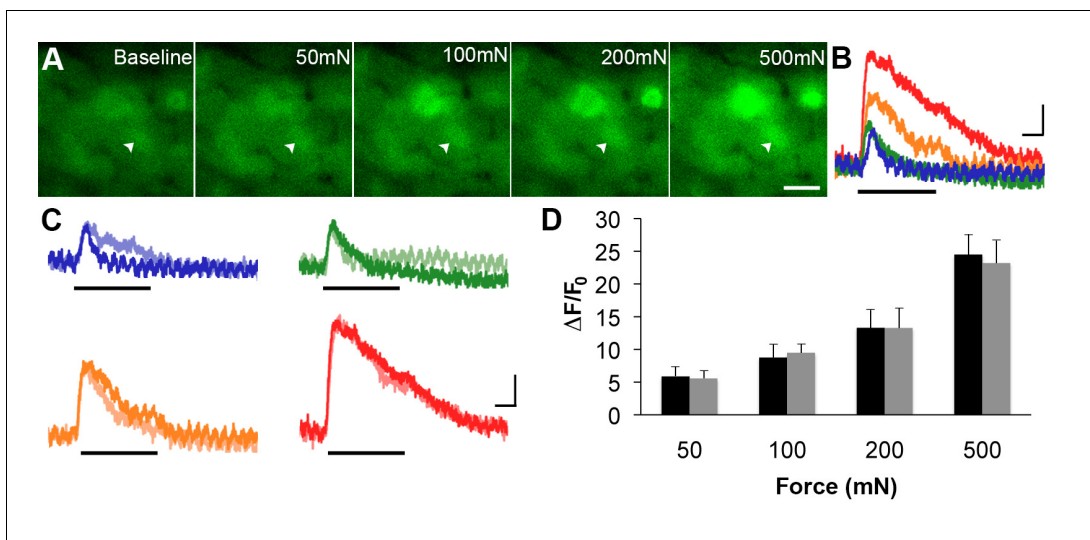

**Figure 6.** GCaMP3 responses to controlled mechanical stimulation encode force intensity and are stable over timein vivo. (**A**) Images of GCaMP3 signal in a mechanically sensitive cell (white arrowheads) at baseline and in response to an ascending series of mechanical forces (left to right). Scale bar, 20 μm. (**B**) Superimposed GCaMP3 traces from the DRG neuron identified in A in response to 50 (blue), 100 (green), 200 (orange), and 500 (red) mN mechanical stimulation (indicated by black bar). Peak GCaMP3 signal increases as the force increases. Scale bar, 1 s; 10 ΔF/F₀. (**C**) Baseline responses to each force of mechanical stimulation (designated color) are superimposed with responses to the same set of mechanical stimuli 30 min later (lighter shade of designated color). Scale bar, 1 s; 10 ΔF/F₀. (**D**) Average GCaMP3 responses to mechanical stimulation at 50, 100, 200 and 500 mN (n=3 mice; n=32 cells). Baseline responses (black) are not significantly different from responses 30 min later (gray) across all forces. Data are represented as mean ± SEM. See *Figure 6—figure supplement 1* for image of mechanical stimulation set-up.

The following figure supplement is available for figure 6:

**Figure supplement 1.** Setup for the application of mechanical stimuli to the skin.

*Xu et al., 2000*), yet their contribution to inflammatory pain remains unclear. Toward this end, changes in responsiveness to mechanical stimulation were analyzed across populations of cutaneous sensory neurons after infusing a cocktail of inflammatory mediators (i.e., IS, consisting of bradykinin triacetate, histamine dihydrochloride, serotonin hydrochloride, and prostaglandin $E_2$, dissolved in carbogen-gassed aCSF and titrated to 6.0 [see *Drug Preparation* for details]) into the skin. Compared to mechanical (i.e. uninjected) and vehicle-injected controls, infusion of IS produced surprisingly diverse changes in sensitivity to mechanical stimuli in sensory neurons across our sample.

Of 45 mechanically sensitive afferents in L6 DRG (n = 5 mice), 23 cells (51%) exhibited no change in GCaMP3 signaling in response to controlled mechanical stimuli following exposure to IS. In contrast, 10 cells (22%) showed significant increases, whereas 12 cells (27%) showed significant decreases in GCaMP3 signal after IS exposure; indeed, some cells lost mechanical sensitivity entirely. This post-IS change in proportions of responsive neurons was dramatically different than those seen in mechanical and vehicle-injected controls (*Figure 7D*, n=6 mice, 83 cells; p=0.004 for increased response, p<0.001 for decreased responses). In cells showing a post-IS increase in GCaMP3 signaling indicative of sensitization, the increase in responsiveness was greatest at the highest force tested (500 mN: p<0.001; post-IS increases at lower forces were evident but not significant, *Figure 7A*). In cells showing decreased mechanical sensitivity post-IS (i.e., desensitization), the change was also most apparent at the highest force (500 mN: p<0.001; *Figure 7B*).

In addition to cells that were mechanically sensitive pre-IS (described above), two newly mechanically sensitive cells appeared de novo following infusion of IS (*Figure 7C*). Exhaustive post-hoc analyses confirmed that these newly emerged cells showed no GCaMP3 signal to mechanical stimulation at any force tested prior to infusion of IS. Importantly, *de novo* responders were not observed in mechanical or vehicle-injected controls at any time point examined (30, 60, or 90 min post-IS). Because new cells would have been expected in these controls if mechanical stimuli were not

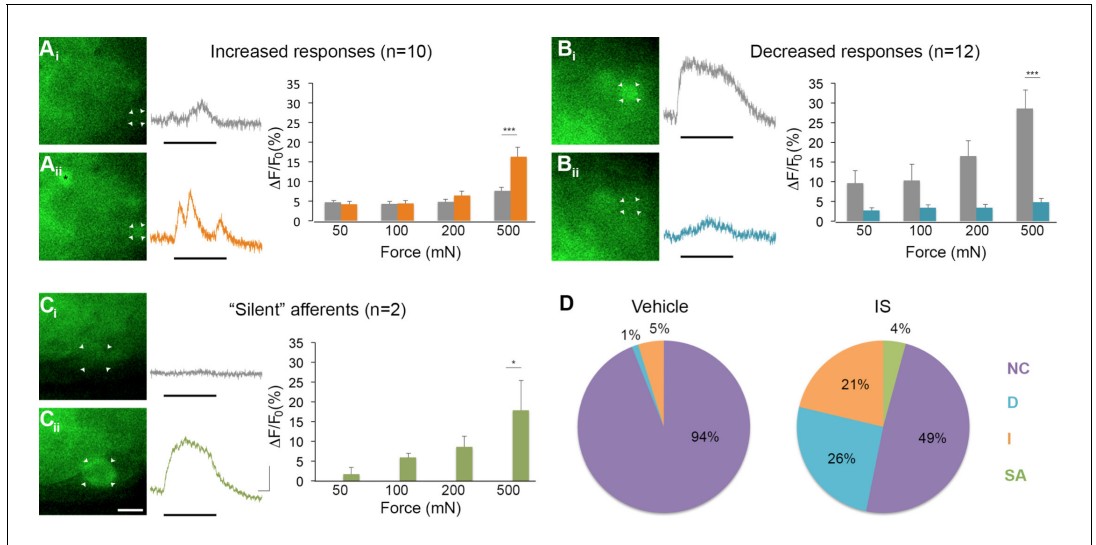

**Figure 7.** Inflammation-induced plasticity in mechanical sensitivity revealed by GCaMP3 in vivo. (A–C) Diverse changes in sensitivity to mechanical stimuli (indicated by black bar under GCaMP3 trace) were revealed when comparing GCaMP3 responses before ($A_i$–$C_i$, gray trace in each panel) and 30 min after ($A_{ii}$–$C_{ii}$, colored trace in each panel) 'inflammatory soup' (IS). These changes are summarized for increased and decreased responses and silent afferents in the graph on the right of each panel. Scale bar, 20 μm; 1 s, 10 $\Delta F/F_0$. Data are represented as mean ± SEM. *p<0.05, **p<0.01, ***p<0.001. (A) Example of an afferent with increased sensitivity to mechanical stimuli after IS ($A_i$–$A_{ii}$; white arrowheads). Black asterisk indicates afferent with increased mechanical sensitivity post-IS within the same visual field. Average post-IS responses were significantly increased only at 500 mN (highest force tested). (B) Example of an afferent with decreased sensitivity to mechanical stimuli after IS ($B_i$–$B_{ii}$; white arrowheads). Average post-IS responses were significantly decreased at 500 mN. (C) Cells that exhibited GCaMP3 signal in response to mechanical stimuli after IS were categorized as 'silent' afferents ($C_i$–$C_{ii}$; example cell indicated by white arrowheads). (D) Summarized results of the proportions of each type of IS-induced change in sensitivity due to vehicle (left; n=6 mice; n=83 cells) and IS (right; n=5 mice; n=47 cells). Compared to changes seen after vehicle, there were significantly more cells that displayed increased and decreased sensitivity to mechanical stimuli. Further, IS caused entirely new sensitivity in a subset of afferents ('silent' afferents), which was never observed in vehicle-injected controls. NC, no change; D, decreased; I, increased; SA, silent afferent.

consistently delivered to the same skin regions over time, the two cells with emergent mechanical sensitivity post-IS clearly fit the definition of 'silent' afferents.

## Properties of 'silent' afferents

To obtain more information on the functional identity of 'silent' afferents before IS exposure, a separate series of experiments was performed wherein radiant heat was briefly applied (3 s) to the skin after recording baseline responses to mechanical stimulation but before IS infusion (n=5 mice, 39 cells). Surprisingly, briefly heating the skin before infusing IS in these experiments was found to recruit even greater numbers of 'silent' afferents than seen in experiments using IS alone, accounting for 21% of the post-IS changes observed (n=8; p=0.018, Chi-square; *Figure 8*). By contrast, the proportion of mechanically sensitive cells showing increased and decreased responses to mechanical stimuli post-IS remained essentially unchanged (increased: n=6, 15%, p=0.473; decreased: n=14, 36%, p=0.299). This unexpected effect of heat on the conversion of 'silent' afferents appeared to depend on IS, since no 'silent' afferents were detected in a separate series of controls that received heat stimuli only (i.e., without subsequent IS; n=3 mice). These findings suggest, therefore, that brief noxious heat exposure may be necessary for the conversion of 'silent' afferents in the skin, but is not sufficient by itself.

Interestingly, 50% (4/8) of all 'silent' afferents identified in this set of experiments responded to heat prior to IS exposure. However, it should be noted that heat was applied to a folded-over flap

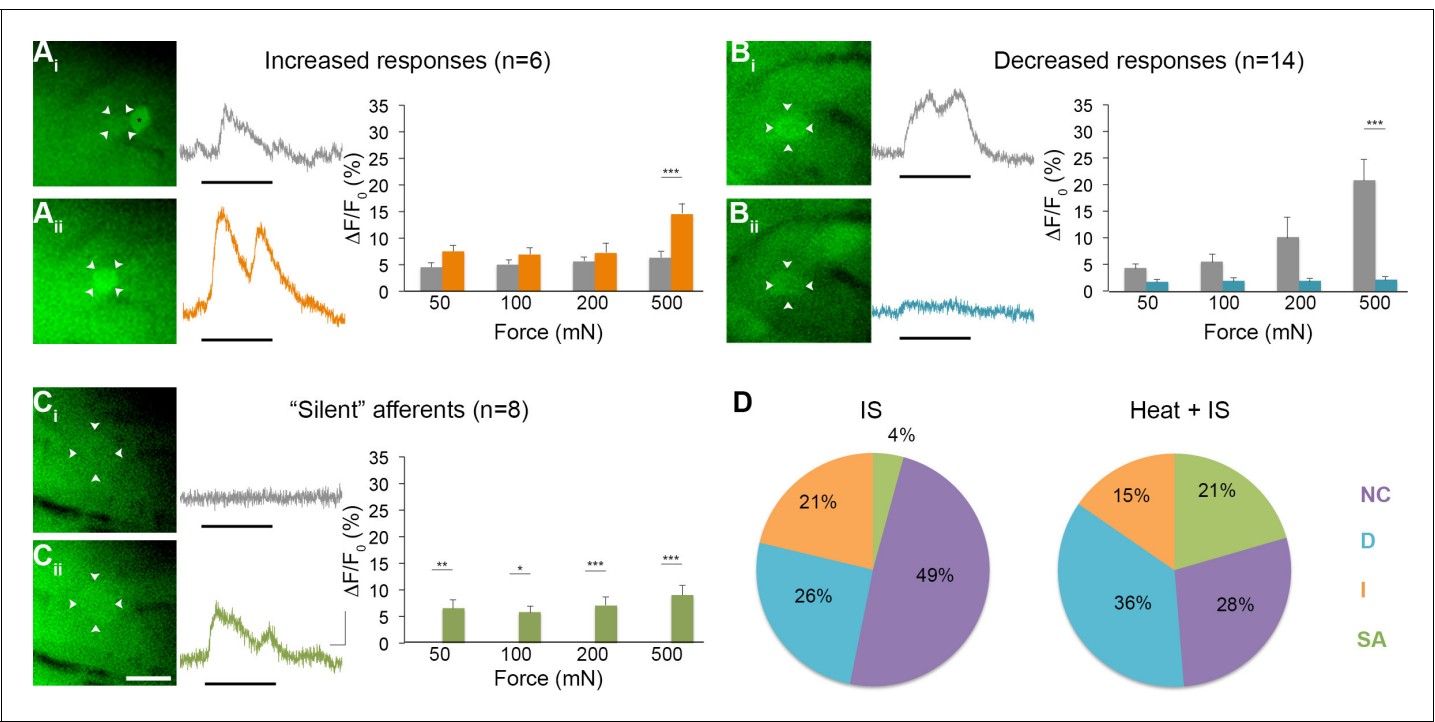

**Figure 8.** Effects of heat exposure to inflammation-induced plasticity in mechanical sensitivity. (A–C) Changes in sensitivity to mechanical stimuli (indicated by black bar) were revealed when comparing GCaMP3 responses before ($A_i$–$C_i$, gray trace in each panel) and 30 min after ($A_{ii}$–$C_{ii}$, colored trace in each panel) heat exposure prior to IS. These changes are summarized for increased and decreased responses and 'silent' afferents in the graph on the right of each panel. Scale bar, 20 µm; 1 s, 10 $\Delta F/F_0$. Data are represented as mean ± SEM. *p<0.05, **p<0.01, ***p<0.001. (A) Example of an afferent with increased sensitivity to mechanical stimuli after IS ($A_i$–$A_{ii}$; white arrowheads). Average post-IS responses were significantly increased only at 500 mN (highest force tested). Black asterisk indicates afferent with decreased mechanical sensitivity post-IS within the same visual field. (B) Example of an afferent with decreased sensitivity to mechanical stimuli after IS ($B_i$–$B_{ii}$; white arrowheads). Average post-IS responses were significantly decreased at 500 mN. (C) Significantly more 'silent' afferents developed mechanical sensitivity after IS when brief noxious heat was applied prior to IS infusion ($C_i$–$C_{ii}$; example cell indicated by white arrowheads). Post-IS responses were significantly increased across all forces. (D) Summarized results of the proportions of each type of IS-induced change in sensitivity due to IS alone (left; n=5 mice; n=47 cells) and heat in addition to IS (right; n=5 mice; n=39 cells). Compared to changes seen after IS alone, there were significantly more 'silent' afferents with de novo mechanical sensitivity when heat was applied prior to IS infusion. NC, no change; D, decreased; I, increased; SA, 'silent' afferent.

of skin positioned between a metal support and the computer controlled mechanical stimulator (*Figure 6—figure supplement 1*; see Experimental Procedures) and that thermocouple measurements indicated that only the top layer of skin was heated to the threshold predicted to activate heat-sensitive nociceptors ($46.7 \pm 0.48°C$; n=3); temperatures in the shielded bottom layer remained well below threshold ($36.9 \pm 0.12°C$; n=3). Thus, if it was possible to heat both layers of the tissue to ca. 46°C, all heat-sensitive nociceptors in the tested skin portion would have been activated, potentially yielding results showing that 100% of the 'silent' afferents were mechanically insensitive heat nociceptors that gained mechanical sensitivity only after IS exposure. By contrast, the percentages of heat sensitive cells across populations that were mechanically sensitive pre-IS were lower for those exhibiting increased, decreased and no change in responses post-IS (33%, 14% and 18%, respectively). Taken together, these findings reveal that while IS exposure produced the full range of effects on polymodal nociceptors, it tended to confer mechanical sensitivity upon heat nociceptors that were previously mechanically insensitive.

In evaluating other properties of 'silent' nociceptors, their acute chemical sensitivity during and up to 20 s after IS infusion was examined. Interestingly, only 2 of 10 (20%) 'silent' afferents across our total sample responded directly to IS during this time window, compared to 50%, 31% and 44% of cells that exhibited an increased, decreased, or no change, respectively, in mechanical sensitivity following IS exposure. Hence, unlike heat, activation by inflammatory mediators was not predictive of the conversion of 'silent' afferents.

Overall, converted 'silent' afferents showed responses across all forces tested ($p<0.05$ for all), and de novo mechanical sensitivity post-IS was evident at the lowest forces tested in the majority of neurons examined (88%). It is important to note that the above results were obtained 30 min after IS exposure. However, in a separate series of experiments aimed at narrowing down the time course of these changes, 'silent' afferents (along with cells that showed significantly increased and decreased responsiveness to subsequent mechanical stimuli) were observed as early as 10 min post-IS (n=3 mice, 37 cells, data not shown).

## Discussion

One of the major limitations to our understanding of plasticity in the somatosensory system has been the inability to monitor activity across multiple populations of primary afferents simultaneously using conventional approaches. To overcome this obstacle, transgenic mice that expressed GCaMP3 in all sensory neurons were utilized to help detect changes in activity levels across diverse and potentially rare cell populations in an unbiased manner. Importantly, the experimental preparation used allows for in vivo, anesthesia-free observation of primary afferent responses. Anesthesia has multiple effects on neurons in the somatosensory pathway and its absence makes it more likely that the observed responses are representative of normal neuronal activity. Although GCaMP3 reliably reported action potentials in virtually all DRG neuron classes with small diameter somata, a subset of putative non-nociceptive afferents lacked fluorescent signals regardless of the amount of evoked activity. It appears likely, therefore, that certain neuronal populations may be refractory to study using GECIs due to insufficient calcium entry during action potentials and/or potent internal buffering mechanisms, and thus interpretation of negative findings with GECIs requires caution.

### Nociceptor plasticity and 'silent' afferents

The primary goal in developing mice with ubiquitous GCaMP expression was to evaluate global changes in nociceptor sensitivity following acute inflammation in vivo. Surprisingly, exposure to inflammatory soup (IS) produced rapid and highly divergent effects overall, with pronounced decreases in mechanical sensitivity in addition to anticipated increases evident across diverse populations of nociceptors. In addition, IS conferred mechanical sensitivity upon a few cells that were previously unresponsive to mechanical stimulation and thus displayed hallmarks of 'silent' afferents (*Davis et al., 1993*; *Kress et al., 1992*; *Meyer et al., 1991*; *Schmelz et al., 1994*; *Schmidt et al., 1995*; *Xu et al., 2000*). Most, if not all, 'silent' afferents were mechanically insensitive heat nociceptors before conversion and were not activated during IS infusion. Interestingly, heat exposure prior to IS infusion recruited a far greater number of 'silent' afferents than either IS or heat stimuli alone, indicating that neither stimulus in isolation is sufficient to drive the conversion of large numbers of 'silent' afferents in the skin, but instead that both in combination are required. This unexpected

finding may help explain previous discrepancies surrounding the frequency of cutaneous 'silent' afferents (*Lynn, 1991*); indeed, the very procedure used to identify afferents by testing responses to diverse stimulus modalities before IS appears to bias the outcome. Nevertheless, these findings hold obvious relevance to inflammatory pain associated with thermal injuries.

The mechanism underlying this unexpected role of heat in the conversion of cutaneous 'silent' afferents is unclear. Mechanically insensitive heat nociceptors express TRPV1 (*Lawson et al., 2008*) and release neuropeptides via local axon reflex that are involved in heat-induced vasodilation (*Cavanaugh et al., 2011*; *Magerl and Treede, 1996*; *Rukwied et al., 2007*; *Xu et al., 2010*). However, simply activating TRPV1 in these cells via brief noxious heat application by itself did not convert 'silent' afferents, indicating that heat and local vasodilation play a largely permissive role. Interestingly, the priming effects of heat may be long-lasting as 'silent' afferents were still recruited in preliminary experiments where IS infusion was delayed by up to 90 min after heat application (data not shown). It is conceivable that TRPV1 activation and local release of neuropeptides may initiate a cascade of downstream effectors that combine with the actions of inflammatory mediators to bring about mechanical sensitivity in this population. However, distinguishing between TRPV1-dependent and -independent mechanisms that may act through contributions from local non-neuronal cells (*Baumbauer et al., 2015*) must await additional genetic manipulations. Regardless of mechanism, most 'silent' afferents became responsive to the lowest mechanical intensity examined, a force that was barely perceptible when applied in a similar manner to the thenar webbing of the experimenters' own hands. These GCaMP3 findings strongly suggest, therefore, that these cells could contribute to mechanical allodynia following acute inflammation. More specifically, they may be expected to contribute to cross-modal dysesthesias such as spontaneous, touch-evoked burning pain (or causalgia), as the spinal circuits that normally process information about noxious heat from these cells would now be driven by the slightest touch.

## Shifting inputs

Whereas the conversion of 'silent' afferents was anticipated, the complexity of changes in other neurons was not. Indeed, only 40% of mechanically sensitive afferents remained unchanged in their mechanical sensitivity after IS exposure. Surprisingly, the percentages of afferents that exhibited increased and decreased sensitivity to mechanical stimulation were comparable when 'silent' afferents were taken into account (30%, increased responses plus 'silent' afferents; 30%, decreased and lost responses). Whereas there are several accepted mechanisms that could explain the increase in sensitivity following exposure to IS, an explanation for decreased sensitivity is less obvious. One possibility is that desensitization is simply a more extreme expression of the mechanisms underlying sensitization; that is, the activation/depolarization produced by IS in the distal endings of primary afferents makes them refractory to subsequent mechanical stimulation (e.g., produced by depolarization-induced inactivation of sodium channels). However, there is little evidence in our dataset to suggest that depolarization may account for the decreased mechanical sensitivity as only 31% of depressed cells responded directly to inflammatory soup (compared to 50%, 41%, and 20% of neurons that showed increased, unchanged, or de novo responses to mechanical stimuli, respectively). Therefore, while depolarization-induced desensitization may explain observations in roughly a third of desensitized neurons, for the majority we believe a more likely explanation is the possibility that peripheral terminals instead became hyperpolarized through direct and/or indirect actions of one or more ingredients in the inflammatory soup, presumably through increased potassium conductance. For example, TREK2 channels expressed in many C fibers (*Acosta et al., 2014*), are activated by protons (*McClenaghan et al., 2016*; *Sandoz et al., 2009*). Further, it is well known that other mediators in the soup (e.g., bradykinin, prostaglandin E2, histamine, and serotonin) can stimulate production of nitric oxide in a variety of cell types and may thus indirectly activate ATP-sensitive potassium channels in sensory neurons (*Chi et al., 2007*; *Du et al., 2011*; *Kawano et al., 2009*; *Zoga et al., 2010*). Most studies emphasize the algogenic properties of these various mediators, but the responses to these mediators were surprisingly diverse and poorly predicted subsequent effects on mechanical sensitivity. Thus, it is likely that the combined effects of these mediators on the mechanical sensitivity of any given neuron reflect not only the specific constellation and relative densities of receptor proteins, but also the proximity of these terminals to sources of secondary mediators in the skin.

While unexpected, these observations of both increased and decreased sensitivity across different populations of sensory neurons are nevertheless consistent with a number of preclinical studies

as well as clinical pain disorders in which both pain and sensory loss occur concomitantly. With respect to preclinical studies, Weyer et al. (*Weyer et al., 2016*) found that 8 weeks after CFA-induced inflammation, mechanical sensitivity in mouse C- and Aδ-fiber afferents was reduced. Whereas this study supports the concept that afferent plasticity in response to inflammation can include both increases and decreases in function, it is unlikely that these long-term changes employ similar mechanisms to those responsible for the changes described here in that they were not seen at earlier times points.

Patients with acute onset Complex Regional Pain Syndrome I (CRPS I) often exhibit both hyperalgesia and hypesthesia in overlapping areas of the affected region as indicated by decreased pain thresholds and increased detection thresholds, respectively (*Huge et al., 2008*). Such a seemingly paradoxical phenomenon has also been observed in other pain disorders associated with peripheral inflammation (*Geber et al., 2008*). Thus, while it is well established that inflammation can lead to hyperalgesia through the sensitization of nociceptors to subsequent stimuli (*Bevan and Yeats, 1991*; *Fjallbrant and Iggo, 1961*; *Randich et al., 1997*; *Grigg et al., 1986*; *Schaible and Schmidt, 1988*; *Steen et al., 1992*; *Steranka et al., 1988*), concomitant hypesthesia has received far less attention, but could reflect the diametric effects of peripheral inflammation of different sensory populations as revealed in the present studies. Nevertheless, a novel hypothesis is hereby presented in which desensitization and/or loss of certain types of nociceptive input following inflammation, coupled with the sensitization and gain of other nociceptive inputs, may serve to unmask and/or sharpen incoming signals that are most relevant to inflammatory pain.

## Material and methods

### Animals
Mice for this study were produced by crossing mice containing a floxed GCaMP3 construct within the *Gt(ROSA)26Sor* locus (The Jackson Laboratory, catalogue #014538) with mice that express Cre recombinase under the EIIa (E2a) adenovirus promoter (The Jackson Laboratory, catalogue #003724). This E2a-Cre strain expresses Cre recombinase in the embryo prior to implantation in the uterine wall producing a mosaic pattern that often includes germ cells. Mice were bred until germline expression of GCaMP3 was achieved as determined by 100% transmission of GCaMP3 to all offspring from crosses involving one GCaMP3-positive male and wild-type females. All studies were performed in accordance within guidelines of the Institutional Animal Care and Use Committees at the Universities of Pittsburgh and Wyoming and the National Institutes of Health guidelines for the Care and Use of Laboratory Animals.

### Immunohistochemistry
Following deep anesthesia with avertin, anesthetized WT (n=4) and GCaMP3 (n=4) mice were transcardially perfused with chilled 4% paraformaldehyde. L5-S1 DRG were dissected, cryoprotected overnight in 25% sucrose, embedded in 10% gelatin, cut at 35 μm using a sliding microtome and incubated as floating sections in blocking solution containing 5% normal horse serum and 0.2% Triton X-100 in 0.1 M PB for 1 hr and then incubated overnight at 4°C in rabbit anti-GFP primary antibody (1:1000 in blocking solution; Sigma-Aldrich, St. Louis, MO). Sections were washed in 3× 5 min in 0.1 M PB and incubated for 1 hr at room temperature in goat anti-rabbit Cy3 fluorescent secondary antibody (1:200 in 0.1 M PB; Molecular Probes/Invitrogen Corporation, Carlsbad, CA). Cells were considered to be GCaMP3-positive if the fluorescent signal was five standard deviations above background as measured with NIH IMAGE software.

### In vitro
Adult GCaMP3 mice (n=5) were deeply anesthetized with isoflurane and transcardially perfused with chilled $Ca^{2+}/Mg^{2+}$-free Hank's balanced salt solution (HBSS; Invitrogen, Carlsbad, CA). Bilateral thoracolumbar DRG were dissected into chilled HBSS and enzymatically treated with cysteine, papain, collagenase type II, and dispase type II to facilitate isolation by mechanical trituration. Isolated neurons were plated on poly-d-lysine/laminin-coated coverslips in Dulbecco modified Eagle medium F12 (Invitrogen, Carlsbad, CA) containing 10% fetal bovine serum and antibiotics (penicillin/streptomycin, 50 U/mL). Cells were flooded with media 2 hr later and imaged within 8–10 hr. Coverslips

were mounted on an inverted microscope stage (Olympus Corporation, Tokyo, Japan) and continuously perfused with normal bath solution (in mM: 130 NaCl, 5 KCl, 1.5 $CaCl_2$, 0.9 $MgCl_2$, 20 HEPES, 5.5 glucose, 0.5 $KH_2PO_4$, 0.5 $Na_2HPO_4$, pH 7.4, osmolality 325 mOsm). Perfusion rate (1 ml/min) was controlled with a gravity flow and rapid-switching local perfusion system (Warner Instruments, Hamden, CT). Solutions were maintained at 32°C using a heated stage and in-line heating system (Warner Instruments, Hamden, CT). Firmly attached, refractile cells were identified as regions of interest. Emission data at 510 nm was acquired via camera (ORCA-ER; Hamamatsu Corporation, Middlesex, NJ) at 1 Hz in response to excitation at 488 nm (Lambda DG-4 and 10-B SmartShutter, Sutter Instrument, Novato, CA) and saved to computer using HCImage (Hamamatsu Corporation, Middlesex, NJ). A solution of 50 mM $K^+$ (Sigma-Aldrich, St. Louis, MO) was applied three times with a 10 min inter-stimulus interval. Depolarization-evoked increases in intracellular calcium concentration were measured by calculating $\Delta F/F_0$), where F is the peak fluorescence signal minus background and $F_o$ is the mean fluorescence signal in a baseline period prior to stimulation. The decay time to 50% of peak fluorescence ($T_{50}$) was determined for 274/462 (~60%) responsive neurons.

## Ex vivo

The ex vivocutaneous somatosensory system preparation used in the present studies was modified slightly from that described in detail previously (*Li et al., 2011*; *Woodbury et al., 2001*). Briefly, adult GCaMP3 mice (n=4) were deeply anesthetized via intramuscular injection of ketamine and xylazine (90 and 10 mg/kg, respectively) and perfused transcardially with oxygenated artificial cerebrospinal fluid (aCSF; in mM: 127.0 NaCl, 1.9 KCl, 1.2 KH2PO4, 1.3 MgSO4, 2.4 CaCl2, 26.0 NaHCO3, and 10.0 D-glucose containing 1 mL/L penicillin/streptomycin). The spinal cord, thoracic DRG, dorsal cutaneous nerves (DCN), and spinal nerves on one side were dissected out at room temperature in a recirculating filtered bath of oxygenated aCSF and then slowly warmed to 30–31°C for data collection. Electrical stimuli were delivered to nerves using suction electrodes; in some cases, the dorsal root was transected near the entry zone and also stimulated with a suction electrode to stimulate all DRG somata with axons in the dorsal root. Fluorescence in DRG somata was imaged with an EMCCD camera (Rolera EM-C2, 2X2 binning; QImaging, Surrey, BC, Canada) using the shortest possible exposure times (3–10 ms). Images were continuously captured using iVision-Mac 4.5 (Biovision Technologies, Exton, PA). Neuronal responses were examined to supramaximal stimulation at 0.5, 1, 5, 10, 20, 100, and 200 Hz (100 μs duration pulses for all except 100 Hz and above which used 50 μs). In some experiments, cells were impaled with microelectrodes (n=21) to determine the relationship between action potential shape, conduction velocity and GCaMP3 signal. DRG somata were impaled with quartz micropipettes (150–300 MΩ) containing 20% Neurobiotin (NB, Vector Laboratories, Burlingame, CA) in 1 M potassium acetate to which a small amount of Alexafluor 555 hydrazide was added (Molecular Probes, Eugene, OR; final concentration ~1%) to allow visualization of the electrode tip and intermittent monitoring throughout recording and staining to verify that the intended cell was impaled. Evoked electrophysiological activity was digitized to disk for subsequent off-line analyses using Spike2 (Cambridge Electronic Design Ltd, Cambridge, UK). Peripheral conduction velocity was calculated from spike latency and the distance between stimulating and recording electrodes measured directly along the nerve.

## In vivo

The in vivo mouse preparation used in the present experiments has been described in detail (*Boada and Woodbury, 2007*), with minor modifications described below. Briefly, adult GCaMP3 mice (n=32) were anesthetized with 4–5% isoflurane in oxygen. Mice were intubated, and the cerebral cortex was exposed via craniotomy and aspirated full thickness prior to immobilization with α-Bungarotoxin (Invitrogen) or pancuronium bromide (Sigma-Aldrich); EKG and end-tidal CO2 were monitored and maintained within normal limits throughout all surgical procedures and recording. A dorsal midline incision was made in trunk skin and DRG at levels L6-S1 were exposed by laminectomy. Dessication was prevented by continuous superfusion with oxygenated aCSF flowing through an in-line heater (MPRE8, Cell MicroControls, Norfolk, VA). The spinal column was secured with custom clamps and the preparation then transferred to the stage of an upright microscope (BX51; Olympus, Center Valley, PA) equipped with fluorescence illumination and water immersion objectives. In most experiments, a 20X objective was used to optimize coverage of DRG. Fluorescence in

DRG somata was imaged with an EMCCD camera (Evolve 512, Photometrics or Rolera EM-C$^2$; QImaging, Surrey, BC, Canada) again using short exposure times (3–10 ms) for rapid continuous capture of images using iVision-Mac 4.5 (Biovision Technologies, Exton, PA).

To examine GCaMP3 response properties in vivo, in some control experiments bipolar stimulating electrodes were inserted through the skin and the peripheral processes of sensory neurons were electrically stimulated using an identical range of intensities (1,10, and 100 V), durations (100 and 50 µs), and frequencies (0.5, 1, 5, 10, 20, and 100 Hz) as in ex vivo studies. However, for studies of inflammation-induced plasticity, a small surface wick electrode was used to prevent damage to the skin that would confound results.

In the latter experiments, an electrical search strategy was used to systematically map the L6 dermatome and locate sensory neurons in the L6 DRG for subsequent study (*Peng et al., 1999*). Briefly, short trains of electrical stimuli (3 pulses at 5 Hz delivered ≥2 s apart) were applied throughout the dermatome while monitoring evoked GCaMP3 activity in the DRG. After locating a promising region of skin, the intensity of electrical stimuli was progressively decreased to pinpoint a region innervated by cells in the DRG that were activated by the lowest intensity. This 'electrically defined' receptive field was then marked with ink and gently folded over onto a smooth custom platform (5×5 mm) attached to a feedback-controlled mechanical stimulator (300C, Aurora Scientific, Richmond, BC, Canada) (*Figure 6—figure supplement 1*). The most lateral edges of the underside of this skin flap were glued to the platform so that the electrically defined field was in the center of the platform facing up. A small cylindrical probe (1 mm$^2$ diameter) attached to the mobile arm of the stimulator was centered on the electrically defined field. Because this resulted in a hairpin loop of skin being pinched between probe and platform, mechanical stimuli delivered to the electrically defined field were also translated to the underlying skin layer, effectively doubling the amount of skin receiving controlled mechanical stimuli. To prevent shifting, the probe remained in contact with the skin throughout the experiment so that successive mechanical stimuli were consistently delivered to the same region. This was verified through visual inspection during experiments and by analyzing stability of evoked fluorescence over time across all cells in control animals. To enable subcutaneous infusion of solutions into the mechanically stimulated skin without movement artifact during imaging, a small, indwelling cannula (32 ga) was secured inside the hairpin loop, adjacent to the line of applied forces. Fluorescence signals in DRG somata were then recorded in response to an ascending series of forces (50, 100, 200, and 500 mN) under baseline conditions; at minimum, 3 min elapsed between each successive stimulus in the series to minimize interaction (*Slugg et al., 2000*). This same series of stimuli was subsequently repeated in the same region 30 min after subcutaneous delivery of 20 µl of 'inflammatory soup' (IS) (n=5) or vehicle (n=6); GCaMP3 responses during IS and vehicle infusions were also recorded. In some experiments, heat was applied before the infusion of IS (n=5) or vehicle (n=3) (average of 23 min prior to IS; median 9; range 5–77 min). In order to ensure mechanical stimulation of the same area, the skin was not removed from the mechanical probe when radiant heat was applied. Because this limited the ability to provide precisely controlled temperatures, a routine procedure was designed that consistently heated the top layer of skin to 46.7 ± 0.48°C (*Figure 6—figure supplement 1*) while heating the bottom layer to a lesser extent (36.9 ± 0.12°C). Using this procedure, the heat sensitivity of cells innervating the top layer of skin (roughly 50% of the sample population) could be determined before and after IS in a binary fashion (i.e., yes or no). Additionally, in three animals, IS-induced changes in mechanical sensitivity were monitored at 10-min intervals for up to 60 min, and up to 90 min in two of these animals. It should be noted that this same series of forces, when applied to the thenar webbing of two of the authors (KMSE, CJW) by pinching with this same cylindrical probe, produced the perception of light to moderate pressure but was not painful.

## Drug preparation

For preparation of IS, bradykinin triacetate, histamine dihydrochloride, serotonin hydrochloride, and prostaglandin E$_2$ (all from Sigma-Aldrich) were dissolved in carbogen-gassed aCSF at concentrations of 10$^{-5}$ M, the pH was titrated to 6.0 with HCl and K$^+$ was increased to 7 mM (*Kessler et al., 1992*; *Steen et al., 1992*). This solution was warmed to 37°C and 20 µl was delivered subcutaneously via the cannula into the electrically defined innervation area. Vehicle was carbogen-gassed aCSF, pH 7.4 at 37°C.

## Data analysis

For in vitro calcium imaging, circular regions of interest (ROIs) were drawn around dissociated neurons using HCImage, and GCaMP3 fluorescence intensity was quantified as $\Delta F/F_0$, where F is the peak fluorescent signal minus background and $F_0$ is the mean fluorescence signal minus background in a baseline period prior to stimulation. Data are presented as mean ± SEM. $\Delta F/F_0$ and the decay time to 50% of peak fluorescence ($T_{50}$) were analyzed by one-way repeated measures ANOVA followed by Tukey's post-hoc test for multiple comparisons.

For ex vivo and in vivo calcium imaging, circular ROIs were drawn around cell bodies using NIH ImageJ, and GCaMP3 fluorescence intensity profiles throughout time-series image stacks (i.e., movie files) were collected. Responses to stimuli were measured by calculating $\Delta F/F_0$ [% = $((F - F_0)/F_0)$ x 100], where F is the peak fluorescence signal and $F_0$ is the mean fluorescence signal in a baseline period prior to stimulation. Because we were not using confocal or two-photon imaging it was possible that some of the detected GCaMP signal for a given ROI could be contaminated by emission from nearby, out-of-focus cells. To control for this concern and to validate measurements obtained from ROIs over specific cells, we also analyzed the potential for spurious signals in nearby non-responder cells by drawing similar-sized ROIs immediately adjacent to each analyzed cell. This allowed us to determine if there was appreciable scatter emanating from the cell of interest and whether any of the observed fluorescence might be coming from out-of-focus cells. This was a bigger concern in cases where we saw an increase in GCaMP3 signal compared to those cases where the signal was reduced or lost. In the vast majority of cases (89%), fluorescent signal was not detectable in ROIs adjacent to analyzed cells. However, to minimize the potential for problems stemming from out-of-focus sources, all responses from analyzed cells of interest were normalized to the signal from adjacent ROIs over non-responding cells, confirming that any reported change in F was not due to focal issues and was specific to the cell of interest.

To assess normal changes in GCaMP3 signal intensity over time in vivo, the signal intensities from single action potentials, evoked via brief trains of electrical stimuli in the periphery at 0.5 Hz to prevent summation (Results), were compared in the same cells at 30 min intervals for up to 90 min (the longest time examined) in uninjected control animals (n=2); to make these data on GCaMP3 signal degradation more directly comparable to data obtained from natural stimulation experiments where multiple imaging sessions were required to complete each series (below), DRGs were intentionally exposed to a cumulative total of ~270 s of illumination interposed between image capture at each 30 min interval. To assess normal changes over time in GCaMP3 signal intensity to natural stimuli (where the number of spikes is not known), controls were performed in naive animals (n=3) by comparing the responses in individual neurons to ascending series of controlled mechanical stimuli repeated at 30 min intervals. Signals were quantified for each cell across the entire population of neurons and compared across time points for calculation of percentage change over time. Confidence intervals from these data from naive, uninjected controls were then used as a normal baseline to evaluate potentially significant changes in experimental animals (vehicle- and IS-injected); changes that fell outside 2 standard deviations of the mean were considered significant. Statistical tests were performed with MiniTab (State College, PA). Descriptive statistics are expressed as mean ± SD when describing heterotypic populations, and mean ± SEM for monotypic entities. Observed differences between cell categories from vehicle- and IS-injected animals were evaluated using Chi-square, whereas IS-induced changes in responses were evaluated with ANOVA with Tukey's post-hoc comparisons. For comparing properties of GCaMP3-responding and non-responding neurons in ex vivo studies, independent sample $t$-tests were used to determine significance (p<0.05).

## Acknowledgements

The authors would like to acknowledge excellent technical support provided by Christopher Sullivan. This work was supported by NS31826 (BMD), DK101681 (JJD), NS044094 (CJW), and P30RR032128 (UW Neuroscience Center; F Flynn, Director).

# Additional information

## Funding

| Funder | Grant reference number | Author |
| --- | --- | --- |
| National Institutes of Health | NS31826 | Brian M Davis |
| National Institutes of Health | DK101681 | Jennifer J DeBerry |
| National Institutes of Health | NS044094 | Jeffery C Woodbury |
| National Institutes of Health | RR032128 | Jeffery C Woodbury |

The funders had no role in study design, data collection and interpretation, or the decision to submit the work for publication.

## Author contributions
KMS-E, JJD, JLS, BMD, CJW, Conception and design, Acquisition of data, Analysis and interpretation of data, Drafting or revising the article

## Author ORCIDs
Jami L Saloman, http://orcid.org/0000-0001-6093-6511
Brian M Davis, http://orcid.org/0000-0002-4646-0569

## Ethics
Animal experimentation: All studies were performed in accordance within guidelines of the Institutional Animal Care and Use Committees at the Universities of Pittsburgh and Wyoming and the National Institutes of Health guidelines for the Care and Use of Laboratory Animals. Approved animal protocol numbers include Univ. of Wyoming IACUC protocol #20131206JW00049-03 (for in vivo studies), # p20131203JW00048-3-03 (for ex vivo studies) and University of Pittsburgh IACUC protocol # 15106942 (for calcium imaging studies).

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
