## [Decision Letter]

Thank you for submitting your article "Profound Alteration in Cutaneous Primary Afferent Activity Following Acute Inflammation" for consideration by *eLife*. Your article has been favorably evaluated by Eve Marder (Senior Editor) and three reviewers, one of whom, David Ginty, is a member of our Board of Reviewing Editors. One of the experts involved in review of your submission, Bruce P Bean (Reviewer #3)has agreed to reveal his identity.

The reviewers have discussed the reviews with one another and the Reviewing Editor has drafted this decision to help you prepare a revised submission.

Summary:

Peripheral neurons can be sensitized to mechanical stimuli through injury, inflammation, or during the course of treatment with certain drugs, and the mechanism underlying this sensitization is incompletely understood. Here, the authors use calcium imaging of DRG neurons to catalog changes in peripheral excitability after experimental induction of an inflammatory response in vivo. Previous studies of how the activity of primary sensory neurons is altered by inflammation have been based on single neuron recordings, so examining simultaneously the changes in many individual neurons is a significant advance. The results are interesting and with a major unexpected finding – that although some neurons have enhanced activity after injection of the inflammatory agents, even more have reduced activity. This is in contrast to prior work reporting only increases in low threshold mechanosensory responses following inflammation (Xu et al. J. Physio. 2000, Ma et al. J. Neurophysiology 2006, Zhu et al. J. Neurophysiology 2015).

The study breaks exciting new ground in the description of silent afferents that are potentiated by heat. It also reports the intriguing observation of afferents that decrease their responses after inflammatory challenge, but does not determine their identity. Overall, the manuscript describes work that anyone interested in the mechanism of pain produced by inflammation will be keenly interested in, and the data set is large enough that the reader gets a clear picture of what happens, at least within the experimental paradigm used by the authors. The paper is fun to read because of the novelty and scope of the measurements and because of the unexpected results. The preparation is elegant and technically innovative yet somewhat hampered by issues of cellular resolution.

1) One surprising result is that a significant number of the cell bodies failed to give calcium signals in response to stimulation of peripheral nerves or the dorsal root, even though electrical recordings showed that they fired action potentials. The authors suggest that this suggests either that these cell bodies do not express voltage-dependent calcium channels or that the action potentials are too narrow to activate them. Both would be very surprising, because in electrophysiological measurements, all DRG neuron cell bodies have large calcium currents, and neurons with far narrower action potentials (e.g. many GABAergic central neurons) still have spike-evoked calcium entry. The authors make a puzzling statement to explain the result " Many medium- to large-sized DRG neurons exhibit narrow APs that have been proposed to lack significant calcium entry (Koerber, 1992; Lu et al., 2006), which may account for the lack of GCaMP3 signal."

The Koerber citation (a chapter in a book) may not be appropriate because it is not a primary research paper, and the Lu et al. paper does not support the assertion – it shows large high-voltage-activated calcium currents in all neurons examined, and all neurons also showed Ca transients measured with fura-2 in response to high K stimulation. There is no reason from this paper to expect that there is a population of medium or large diameter that would fail to show detectable Ca transients evoked by trains of action potentials. The authors fail to discuss two possibilities – that some cells simply do not express functional GCaMP at high enough levels to respond to action potentials or that there are differences in calcium buffering between different cell types. In Figure 1, it looks as there are a number of large cells that show little or no immune signal with the anti-GFP antibody. This point should be further discussed in a revision.

2) There is an over-interpretation of the data implied in the title. The authors do not actually study inflammation, but the effects of a limited cocktail of agents that are involved in inflammation. (It would be good if the make-up of the inflammatory soup were given in Results when it is first used – the reader has to flip to Methods to get this important information.) The inflammatory soup is similar to that in other papers, but it is still questionable how well it mimics natural inflammation. This is reinforced by the authors' interesting finding that the number of previously silent afferents becoming active is greatly increased by briefly heating the skin prior to injection of the soup. The effects of heating the skin may be mediated through TRPV1 receptors, which suggests that its effects might be mimicked by adding to the soup some agents capable of more robust TRPV1 stimulation…in other words, a change in recipe in the soup used to mimic inflammation might significantly change the results. This should be discussed, and a title such as "Profound Alteration in Cutaneous Primary Afferent Activity Produced by Inflammatory Mediators" would more accurately describe the findings.

3) The authors do not offer potential explanations for the neurons that become less active after exposure to the mediators. One that might be mentioned is that the neurons become so strongly depolarized that their sodium channels become inactivated. This should be mentioned.

4) Figure 5 introduces in vivo DRG imaging, which raises a technical concern with the study. Neurons in sensory ganglia are tightly packed, and related work featuring population of imaging of sensory ganglia (Barreto et al. Nature 2015, Kim et al. Neuron 2014, William, Chang, Strolich et al. Cell 2016) use confocal or multiphoton microscopy to achieve signal isolation between adjacent cells. In panels A-D, it is difficult to make out cell bodies. Out-of-focus fluorescence will produce changes to dF/F if adjacent neurons become spontaneously active (F increases), which is reported to occur in response to an inflammatory stimulus (Xu et al. J. Physio. 2000, Ma et al. J. Neurophysiology 2006). The authors should address this technical issue.

5) Cheryl Stucky's group recently reported that inflammation causes reduction in mechanical sensitivity in aged mice, but not in young mice. The authors should discuss how their findings are related to the phenomenon observed by Stucky and her colleagues.

---

## [Author Response]

[…]

*1) One surprising result is that a significant number of the cell bodies failed to give calcium signals in response to stimulation of peripheral nerves or the dorsal root, even though electrical recordings showed that they fired action potentials. The authors suggest that this suggests either that these cell bodies do not express voltage-dependent calcium channels or that the action potentials are too narrow to activate them. Both would be very surprising, because in electrophysiological measurements, all DRG neuron cell bodies have large calcium currents, and neurons with far narrower action potentials (e.g. many GABAergic central neurons) still have spike-evoked calcium entry. The authors make a puzzling statement to explain the result " Many medium- to large-sized DRG neurons exhibit narrow APs that have been proposed to lack significant calcium entry (Koerber, 1992; Lu et al., 2006), which may account for the lack of GCaMP3 signal."*

*The Koerber citation (a chapter in a book) may not be appropriate because it is not a primary research paper, and the Lu et al. paper does not support the assertion – it shows large high-voltage-activated calcium currents in all neurons examined, and all neurons also showed Ca transients measured with fura-2 in response to high K stimulation. There is no reason from this paper to expect that there is a population of medium or large diameter that would fail to show detectable Ca transients evoked by trains of action potentials. The authors fail to discuss two possibilities – that some cells simply do not express functional GCaMP at high enough levels to respond to action potentials or that there are differences in calcium buffering between different cell types. In Figure 1, it looks as there are a number of large cells that show little or no immune signal with the anti-GFP antibody. This point should be further discussed in a revision.*

The reviewer raises an important issue and correctly notes that Lu et al. (2006, J. Neurophys. 577.1:169) found both calcium transients (using Fura-2 imaging) and calcium currents (using whole cell patch clamp) in dissociated rat DRG neurons of all sizes, including neurons >40µm in diameter. Figure 2 and Figure 3 in Lu et al., 2006 show that for calcium transients in dissociated rat DRG neurons, the largest and most prolonged transients are in small cells and the smallest and shortest are in large cells (this pattern is true whether they used high [K^+^] or 4 seconds of current injection to depolarize the cell). In contrast, in whole cell voltage clamp, the greatest current estimates are seen in the reverse order (Figure 4). In other words, despite seeing small Ca^2+^ transients in large cells, I-V curves for either high-voltage activated Ca^2+^ currents or low-voltage activated Ca^2+^ currents (T-type seen in 4-16% of the different populations) was greater in larger cells, exactly opposite of what would have been predicted by looking at the Ca^2+^ transients alone. This indicates that other processes besides channel density (which may be higher in large cells), such as calcium buffering and reuptake, dominate regulation of free intracellular Ca^2+^ in response to activation. Thus, the results in the present paper are not completely discordant with the Lu et al. results. Lu et al. reported the smallest Ca^2+^ transients in large cells whereas we saw large cells in which we could not elicit observable Ca^2+^ signals as detected by GCaMP3. The difference between seeing small transients (Lu et al.) and no signal at all (our paper) could in part be due to the higher K_d_ for Ca^2+^ of GCaMP3 relative to Fura-2 (350nM vs. 140nM). However, in transgenic mice ubiquitously expressing GCaMP6s (K_d_ = 144nM), no signal could be detected in many large diameter DRG neurons during dorsal root stimulation at 100 Hz in ex vivo preparations or in spindle afferents during maintained muscle stretch (in vivo preparations; Smith-Edwards and Woodbury, unpublished data), another population of large cells with narrow spikes that have been identified to express high levels of parvalbumin (a potent calcium binding protein). While it is also possible that the difference between large and small cells is due to insufficient expression of GCaMP in large cells, it is not clear why the lack of expression would be restricted to cells of a certain diameter since these large cells are functionally heterogeneous and GCamP expression is driven by a cell type independent CAG hybrid promoter in the ROSA26 locus. Finally, it should be noted that we only observed a lack of a GCaMP3 signal in 2.5% of all neurons in culture, a small minority. The reason this issue was raised originally was the realization that investigators using viral expression would normally interpret cells that do not exhibit calcium transients following infection with GCaMP vectors as either “non-responders” or worse, not part of the circuit, whereas our data, based on germline expression, raised the possibility that some cells could express GCaMP3 protein but produce false negatives due to insufficient levels of free calcium (either because of smaller transients or powerful sequestration) following activation. Parts of this discussion were added to the text in the last paragraph of the subsection “GCaMP3 signal in DRG ex vivo”.

*2) There is an over-interpretation of the data implied in the title. The authors do not actually study inflammation, but the effects of a limited cocktail of agents that are involved in inflammation. (It would be good if the make-up of the inflammatory soup were given in Results when it is first used – the reader has to flip to Methods to get this important information.) The inflammatory soup is similar to that in other papers, but it is still questionable how well it mimics natural inflammation. This is reinforced by the authors' interesting finding that the number of previously silent afferents becoming active is greatly increased by briefly heating the skin prior to injection of the soup. The effects of heating the skin may be mediated through TRPV1 receptors, which suggests that its effects might be mimicked by adding to the soup some agents capable of more robust TRPV1 stimulation…in other words, a change in recipe in the soup used to mimic inflammation might significantly change the results. This should be discussed, and a title such as "Profound Alteration in Cutaneous Primary Afferent Activity Produced by Inflammatory Mediators" would more accurately describe the findings.*

We agree that the title is overstated and have changed it to reflect more accurately the nature of our experimental manipulation (actually, we liked the reviewer-proposed title and have adopted it for our paper). We have also added the formulation for the soup to the Results section (subsection “Inflammation-induced plasticity”, first paragraph). We did discuss using more powerful sensitizing agents for TRPV1 such as capsaicin. However, only a fraction of cutaneous afferents express TRPV1 (ca. 30%, Malin et al., 2011, J Neurosci. 31:10516) and capsaicin is not an endogenous ligand for TRPV1, i.e., one that would be present in inflamed tissue. That the inflammatory soup was acidic (pH 6.0) was intended to provide protons, one of the most potent endogenous activators of TRPV1 that could activate the channel in a manner that might replicate conditions found naturally in inflamed skin.

*3) The authors do not offer potential explanations for the neurons that become less active after exposure to the mediators. One that might be mentioned is that the neurons become so strongly depolarized that their sodium channels become inactivated. This should be mentioned.*

We thank the reviewer for bringing our attention to this omission. The reviewer provides a reasonable explanation for the observed results, although there is unfortunately little evidence in our dataset to suggest that depolarization may account for the decreased mechanical sensitivity in this population overall, as only 31% of these cells responded directly to inflammatory soup (compared to 50%, 41%, and 20% of neurons that showed increased, unchanged, or de novo responses to mechanical stimuli after exposure to mediators, as reported). Therefore, while depolarization-induced inactivation of sodium channels may explain observations in roughly a third of desensitized neurons, for the majority we believe a more likely explanation is the possibility that peripheral terminals became hyperpolarized through direct and/or indirect actions of one or more ingredients in the inflammatory soup, presumably through increased potassium conductance. For example, TREK2 channels, expressed in many C-fibers (Acosta et al., 2014, J Neurosci 34:1494), are activated by protons (Sandoz et al., 2009, Proc Natl Acad Sci106:14628; McClenaghan et al., 2016, J Gen Physiol, 147:497). Further, it is well known that other mediators in the soup (e.g., bradykinin, prostaglandin E2, histamine, and serotonin) can stimulate production of nitric oxide in a variety of cell types and may thus indirectly activate ATP-sensitive potassium channels in sensory neurons (Chi et al., 2007, Brain Res. 1145:28; Kawano et al., 2009, Mol Pain 5:12; Zoga et al., 2010, Mol Pain, 6:6; Du et al., 2011, Mol Pain 7:35). Most studies emphasize the algogenic properties of these various mediators, but the responses to these mediators were surprisingly diverse and poorly predicted subsequent effects on mechanical sensitivity. Thus, it is likely that the combined effects of these mediators on the mechanical sensitivity of any given neuron reflect not only the specific constellation and relative densities of receptor proteins but also the proximity of these terminals to sources of secondary mediators in the skin. Portions of the above were added to the Discussion (subsection “Shifting inputs”, first paragraph).

*4) Figure 5 introduces* in vivo *DRG imaging, which raises a technical concern with the study. Neurons in sensory ganglia are tightly packed, and related work featuring population of imaging of sensory ganglia (Barreto et al. Nature 2015, Kim et al. Neuron 2014, William, Chang, Strolich et al. Cell 2016) use confocal or multiphoton microscopy to achieve signal isolation between adjacent cells. In panels A-D, it is difficult to make out cell bodies. Out-of-focus fluorescence will produce changes to dF/F if adjacent neurons become spontaneously active (F increases), which is reported to occur in response to an inflammatory stimulus (Xu et al. J. Physio. 2000, Ma et al. J. Neurophysiology 2006). The authors should address this technical issue.*

We were also concerned with the issue of out-of-focus fluorescence raised by the reviewer. Our goal was to capture maximal temporal information about changes in neural activity occurring simultaneously at scattered locations throughout the DRG. While there is no doubt that greater spatial resolution is possible, especially using multi-photon technology combined with a resonant scanner, we did not have access to such equipment. Our experimental approach (widefield imaging at millisecond timescales with medium-high magnification objectives) was extremely well suited to capture rapid events taking place simultaneously at widely scattered locations throughout the DRG, with the trade-off that this was also prone to the out-of-focus concerns. To help minimize these issues and to validate measurements obtained from regions of interest (ROIs) over specific cells, we also analyzed the potential for spurious signals in nearby non-responder cells by drawing similar-sized ROIs immediately adjacent to each analyzed cell. This allowed us to determine if there was appreciable scatter emanating from the cell of interest and whether any of the observed fluorescence might be coming from out-of-focus cells. This was a bigger concern in cases where we saw an increase in GCaMP3 signal compared to those cases where the signal was reduced or lost. In the vast majority of cases (89%), fluorescent signal was not detectable in ROIs adjacent to analyzed cells. Importantly, however, to minimize the potential for problems stemming from out-of-focus sources, all responses from analyzed cells of interest were normalized to the signal from adjacent ROIs over non-responding cells, confirming that any reported change in F was not due to focal issues and was specific to the cell of interest. We thank you for bringing it to our attention that these details were not included in the Methods section. Please note the additions in the second paragraph of the subsection “Data analysis” that properly address this concern.

With respect to the reviewer’s specific concerns surrounding spontaneously active (SA) cells, there is only minor evidence of IS-induced spontaneous activity in our dataset and little evidence that it influenced our analyses, results, or conclusions. Spontaneous activity was occasionally observed in control experiments (i.e., no IS). Of the 13 experiments involving IS infusion, SA cells were only observed in 7 (54%) after IS exposure and at low numbers (only 2 SA cells in 4 of the cases and only 1 in each of the other 3 cases). Of the 11 SA cells post-IS, 2 exhibited spontaneous activity prior to IS exposure, 4 were mechanically sensitive and therefore included in our analyses (including 1 that displayed spontaneous activity pre- and post-IS and whose response to mechanical stimuli was unchanged following IS exposure). None of the remaining 6 cells were located near cells that were analyzed (the closest was at least 2 cell diameters away from an analyzed cell and its signals were not detectable in the ROI over the cell of interest). Moreover, in all 11 SA cells, ongoing discharge frequencies were low (from ~0.5-2 Hz) throughout the period of mechanical sensitivity testing; one cell gave brief yet infrequent high-frequency bursts (fluorescent signals lasting up to 1.4 seconds, repeated every 9-10 sec) shortly after IS exposure, but by 12 minutes this bursting activity had been reduced to an irregular ongoing discharge (~0.5 Hz) that persisted throughout mechanical sensitivity testing. (This cell did not respond to mechanical stimulation and thus, was not among those included in analyses.) The infrequency of IS-induced spontaneous activity seen here was foreshadowed by findings from Ma et al. (2006, J Neurophysiol); in that study, only 2 /113 (1.8%) sensory neurons from control DRG responded to IS; by contrast, the numbers of IS-responsive cells were much higher in DRG that had undergone chronic compression.

*5) Cheryl Stucky's group recently reported that inflammation causes reduction in mechanical sensitivity in aged mice, but not in young mice. The authors should discuss how their findings are related to the phenomenon observed by Stucky and her colleagues.*

Weyer et al. (eNeuro 2016; 10.1523/ENEURO.0115-15.2015) found increased mechanical sensitivity in C-fibers after “acute” inflammation in young mice but decreased mechanical sensitivity in both C- and Aδ-fibers after “chronic” inflammation in both young and aged mice using single-fiber recordings. In some respects, these findings appear to parallel the present results, although it is important to note numerous differences. Weyer et al. used Complete Freund’s Adjuvant (CFA) to generate inflammation and performed experiments days after injection to compare effects on sensory neurons during acute (2 days) versus chronic (8 weeks) phases of CFA-induced inflammation, whereas the present study analyzed far more rapid (i.e., immediate acute) changes following infusion of a defined cocktail of inflammatory mediators (inflammatory soup, or IS); indeed, post-IS responses were measured 30 minutes after the infusion of inflammatory mediators (and in some cases as early as 10 minutes after exposure). A couple of similarities are noteworthy, including the fact that increased responses to suprathreshold mechanical forces were observed in both studies, a finding that has been reported in multiple prior reports. Further, while decreased responses were found in both studies, Weyer et al. did not observe decreased responses until 8 weeks post-CFA, whereas the present study found this after just 10 minutes post-IS. Because of the widespread differences in methodologies, including the manner in which inflammatory environments were created, the time course for measuring changes, and the differences used to measure changes (e.g., single-fiber recordings in Weyer et al. versus simultaneous population-based imaging used presently), it is difficult to speculate upon possible underlying mechanisms that may be shared between the two studies. However, we agree that referencing the findings of Weyer et al. would benefit the manuscript, and this addition can be found in the second paragraph of the subsection “Shifting inputs”.